# Medical students' perceptions of their preparedness to care for LGBT patients in Taiwan: Is medical education keeping up with social progress?

Peih-Ying Lu[1,2,3]*, Anna Shan Chun Hsu[1], Alexander Green[4], Jer-Chia Tsai[1,3,5]

1 College of Medicine, Kaohsiung Medical University, Kaohsiung, Taiwan, 2 College of Humanities and Social Sciences, Kaohsiung Medical University, Kaohsiung, Taiwan, 3 Center for Medical Education and Humanizing Health Professional Education, Kaohsiung Medical University, Kaohsiung, Taiwan, 4 Harvard Medical School, Massachusetts General Hospital, Boston, MA, United States of America, 5 Department of Internal Medicine, Kaohsiung Municipal Ta-Tung Hospital, Kaohsiung, Taiwan

* peyilu@kmu.edu.tw

**Data Availability Statement:** We are unable to share the transcripts from the focus groups and interviews as these contain potentially identifying or private information. Participants were not

## Abstract

### Introduction

Integrating training on health equity of sexual and gender minorities (SGM) in medical education has been challenging globally despite emphasis on the need for medical students to develop competence to provide adequate care for diverse patient groups. This study elicits Taiwanese medical students' perceptions of their values and preparedness to care for Lesbian, Gay, Bisexual, or Transgender (LGBT) patients using a qualitative approach that considers broader societal changes, and more focused topics such as the provision of relevant training in medical education.

### Methods

Eighty-nine medical students/trainees from two southern Taiwanese medical schools (one public and one private) participated in focus groups (n = 70) and individual interviews (n = 19). Qualitative analysis was conducted using inductive thematic analysis.

### Results

Participants (i) expressed wide social acceptance and openness toward LGBT individuals, but were unsure of ways to communicate with LGBT patients; (ii) confirmed that stigmatization and biases might be developed during their training; (iii) recognized gender stereotypes could have negative impacts on clinical reasoning; (iv) considered themselves prepared to care for LGBT patients, yet equated non-discriminatory attitudes to preparedness; (v) acknowledged a lack of relevant professional skills; (vi) implicated curriculum did not address LGBT issues systematically and explicitly.

informed and did not consent for the data to be shared publicly. This project was reviewed and approved by the National Cheng Kung University Governance Framework for Human Research Ethics. They can be contacted by email at em51020@email.ncku.edu.tw. However, we are able to share our NVivo codebook and have uploaded it as a supporting information file.

**Funding:** This work was supported by Taiwan's Ministry of Science and Technology under the following grant: MOST 103-2511-S-037-002-MY2. The funder had no role in study design, data collection and analysis, decision to publish, or preparation of the manuscript.

**Competing interests:** The authors have declared that no competing interests exist.

## Conclusion

This study has identified the insufficiencies of current medical training and inadequate preparedness of medical students/trainees to provide better care for LGBT patients. It provides insights for medical educators to design and implement effective medical curriculum and training, and faculty development programs to equip medical students/trainees with self-awareness and competencies to more readily provide holistic care for SGM, in keeping up with social progress, and promote health equity for a more diverse patient population.

## Introduction

There is an urge for the end of discrimination against sexual and gender minorities (SGM) to ensure equity of health provision [1,2]. Placed in Bronfenbrenner's Ecological systems theory [3], at the macro, social level, equity in health care includes policies and legislation that interacts with cultures and societies at the meso level and individuals at the micro level. Across different age groups, patients who identify as Lesbian, Gay, Bisexual, or Transgender (LGBT) often face challenges and disparities when seeking to satisfy their health care needs [4–8]. Global discussions about the provision of satisfactory medical care for LGBT patients and relevant trainings in medical education were initiated. Over the past few decades, movement to ensure equity of health care provision for LGBT patients has been more common in Western societies [9–13] than in Asian societies, particularly those with more conservative religious beliefs and cultural norms. In some Asian countries, quality of health care for LGBT individuals may be affected by social discrimination and regulations against same-sex relations and non-binary forms of gender expression [14]. In other Asian countries such as Taiwan, discrimination against LGBT individuals may be less explicit, but care could still be impacted by biases related to cultural emphasis on family structure and filial piety, reproduction, community orientation, hierarchical authority, and paternalism [15–17]. Taken together, these factors might influence health equity of LGBT individuals.

In Taiwan, acceptance towards LGBT individuals has been increasing as reflected by more awareness of HIV campaigns, LGBT rights movements and their related NGOs [17], followed by legalization of same sex marriage in 2019. Legal advances and social trends in Taiwan reflected that the younger generation tends to have more open attitudes toward LGBT individuals [18]. With recent shifts, it is now much more likely for health care professionals to care for a population of LGBT individuals who are open to revealing their sexual and gender identities [19]. However, whether these shifts have been amply trickled top-down to transform medical education training to increase medical students'/trainees' understanding of LGBT-related health equity issues needs further investigation.

Many studies on healthcare equity for LGBT individuals have indicated that medical schools in western context face similar challenges in integrating topics relating to gender and sexuality diversity into the curriculum [20–24]. Medical educators need to address these challenges from students' learning environment in formal, informal, and hidden curriculum. Biases and inadequate training within these domains, which may contribute to health disparities in vulnerable populations, are also of concern in medical education programs [25–27].

To date, most of the relevant studies on LGBT related issues in medical education investigated students' attitudes, how comfortable they feel about taking care of the patients [28,29], and curricular design and content [30,31]. Most studies were conducted in western contexts

[22,25,31–37], with only a few in Asia [16,29,38,39] since many Asian countries are still relatively conservative about topics related to LGBT individuals in general [17] and in medical education more specifically [29,39]. A study conducted in Japan [29] emphasized the need for increased education on sexual and gender minorities (SGM) in medical education although they found only roughly one-fourth of medical schools surveyed reporting they provided some kind of education on topics related to these groups. For the rest of the medical schools, the reasons for schools not offering these curricula on SGM included lack of knowledge, preparation, or suitable instructors. In Taiwan, the importance of curricula on health equity of SGM in pre-clinical and clinical training was underscored as one of the criteria for medical education accreditation [40], with modified standards that are mainly based on those of the Liaison Committee on Medical Education (LCME) [41] requiring medical students to develop abilities to recognize and deal appropriately with biases related to gender and sexual orientation. Despite these efforts, whether these issues have been adequately embedded in curricula remain relatively unexplored in this global discourse of health equity.

The current study used qualitative method to examine medical students' preparedness to care for LGBT patients. This study is also part of a larger project that used mixed methods to examine Taiwan medical students' perspectives in caring for patients from diverse groups distinguished by age, ethnicity, level of trust in Chinese/western medicine, and identification as LGBT individuals [42]. Currently, there has been little evidence to indicate how prepared healthcare providers in Taiwan are to care adequately for publicly open LGBT patients. There are also gaps in identifying the influences existing curricula has on shaping students' and learners' understanding and response of how to interact with LGBT patients.

Findings from quantitative data collected by the current research team showed student respondents felt somewhat adequately prepared to provide care for LGBT patients while teachers felt the contrary [42]. In the present study, we specifically address this discrepancy by exploring the perspectives and learning experiences of Taiwanese medical students on caring for LGBT patients, and on the related medical curriculum and training environment, in hopes of expanding the limited research on LGBT related issues in an East Asian medical education context.

## Methods

### Participants

In Taiwan, most medical schools are undergraduate entry level, with years one to four as the pre-medical and pre-clinical stages and years five to six as the clinical stage. All medical graduates attend the compulsory two-year postgraduate (PGY) general medicine training program. Undergraduate medical students/trainees were recruited in two phases. In the first phase, a survey on cross-cultural care competence involving diverse cultural group (Table 1) was administered to 1545 medical students from one public and one private medical school in southern Taiwan and 1120 (72.5%) responses were received.

Initial results from the survey were then analyzed to serve as baseline for qualitative data collection. The researchers recruited from each school a random sample of students from the cohorts of students that completed the survey and conducted at least one focus group (FG) and one interview for each year of study, involving a total of 89 students who participated in 14 FGs (with four to six participants in each session) and 19 individual interviews. Foreign students studying abroad in Taiwan were excluded. For FGs, cohort representatives first posted recruitment messages on class bulletins, then students were recruited through convenience and snowball sampling. Interview participants were recruited through purposive sampling via email or phone. Demographic characteristics of participants in FGs and interviews are shown in Table 2.

**Table 1. Diverse groups included in cross-cultural care competence survey.**

| |
|---|
| 1. Cultures different from your own |
| 2. Health beliefs or practices at odds with Western medicine |
| 3. Distrust the Taiwanese health care system |
| 4. Speak limited Mandarin |
| 5. New immigrants/spouses of new immigrants |
| 6. Religious beliefs affect treatment |
| 7. Use complementary or alternative medicines |
| 8. Minority groups: foreign labors |
| 9. Minority groups: the indigenous |
| 10. Gay, lesbian, or bisexual |
| 11. Transgender |
| 12. Persons with disabilities |

## Design

Qualitative data were triangulated to obtain in-depth descriptions of participants' perceptions and experiences. Researchers conducted most interviews and FGs on medical school campuses or meeting rooms of universities' affiliated hospitals, with one interview conducted at a quiet café close to campus. Structured questions were prepared (See Box 1 for English translation of the questions, which were given in Mandarin), covering general attitudes to students' and trainees' preparedness and competence in caring for diverse groups, including LGBT individuals. Interviews and FGs lasted approximately 40–60 minutes each and were audio-recorded

**Table 2. Demographics of student participants from two medical schools in Taiwan: 2015–2017.**

| | Qualitative (N = 89) | |
|---|---|---|
| **% of Student Respondents** | **Focus Groups (N = 70)** | **Interviews (N = 19)** |
| Gender | | |
| Male | 58.6% | 57.9% |
| Female | 41.4% | 42.1% |
| Year in Medical Program | | |
| 2 | 14.3% | 21.1% |
| 3 | 14.3% | 15.8% |
| 4 | 14.3% | 15.8% |
| 5 | 12.9% | 15.8% |
| 6 | 15.7% | 10.5% |
| 7 | 12.9% | 10.5% |
| PGY | 15.7% | 10.5% |
| Medical Schools | | |
| KMU | 45.7% | 63.2% |
| NCKU | 54.3% | 36.8% |

KMU: Kaohsiung Medical University.

NCKU: National Cheng Kung University.

PGY = Post-Graduate Year.

*Note.* Taiwanese medical schools generally admit high school graduates. Starting from 2013, the original 7-year medical education program changed to 6-years, with the original final internship year moved to postgraduate clinical training (PGY). As the study was conducted in 2015–2017 before the implementation of the two-year PGY program, only the first PGY year was included in the study.

and transcribed verbatim. The same semi-structured questions, which were created based on findings from quantitative data and validated by a panel involving project researchers and two clinicians, were used for both interviews and FGs. We are aware that FGs and interviews are different methods of data collection, however, we believe these differences do not warrant the need to conduct separate sets of analyses as the phenomenon identified by participants in both groups were similar. Thus, these data were triangulated and interpreted together.

---

### Box 1. Questions from student focus groups & interviews

#### Focus Groups

- In your understanding, what is cross-cultural care competence?

- Do you think there are biases and/or stereotypes that are formed during doctors' training in medical education?

- With regards to caring for homosexual, bisexual, transgender individuals, and other groups with gender identities different from heterosexuals, how is your preparedness level?

- Do you think the medical curriculum provides sufficient training on cross-cultural care?

#### Individual Interview

*Structured Questions*(related)

- In your understanding, what is cross-cultural care competence?

- Based on our previous study, students thought that during the process of medical training, doctors (or medical students) can easily form biases. What is your view on this?

- How prepared are you in terms of taking care of LGBT patients? Which courses (including formal and implicit) or experiences do you think help with your preparedness level?

*Follow-up Semi-Structured Questions*

- Rate your current preparedness level (on a scale of 1–10).

- In terms of communication skills, do you already have knowledge of and preparedness for these groups of individuals? For instance, you indicated that LGBT individuals have their own mindset and beliefs. How prepared are you in these aspects?

- Based on your observations, how prepared do you think your peers are for LGBT individuals?

- (follow-up from a previous question) So when you encounter individuals who are of different gender identity, for instance transgenders, you wouldn't have trouble communicating with them?

*Note.* There were other questions that were asked during the focus groups and interviews, but only questions pertinent to the present study are listed; *Follow-up Semi-Structured Questions* were asked to probe for further information based on what respondents had said.

---

Ethics approval for this study was obtained from the National Cheng Kung University Governance Framework for Human Research Ethics (Approval No.: NCKU HREC-E-106-291-2/ NCKU HREC-E-104-035-2). Participants in focus groups and interviews provided written consents at the beginning of the sessions.

## Analysis

Two members of the research team reviewed and discussed each transcript to reach consensus on differences in coding. Inductive thematic analysis was conducted. Transcripts were reviewed, coded manually, and processed using Nvivo 10. Researchers independently reviewed transcribed data and identified recurring topics. From these topics, codes for texts that expressed similar contents were established and students' quotes were sorted according to these codes. Codes were collated into various themes that were then reviewed, defined, and labeled. Only data relevant to LGBT care provision is extracted and discussed. Similar codes were clustered to identify emerging themes and sub-themes. Recurring contents were grouped into four main themes and nine categories. The established themes and categories were translated to English and compared to the original data to ensure they reflect the original contents.

## Results

Although students had varying educational experiences at different stages of medical training at the two schools, they still had remarkably similar perceptions. The account given therefore does not particularly differentiate between different stages. "LGBT" is used to encompass both individuals with diverse sexual orientation or gender identity in this study even though respondents' discussions and examples mainly focused on people with same sex sexual orientation, with little mentioning of bisexual or transgender despite students being asked about individuals who identify as LGBT. The four themes and nine categories are listed in Table 3 and presented in the following sections.

## Attitude and perceptions

**Wide social acceptance and self-perceived openness.** The influences of the change of Taiwan's social climate were recognizable in the ways respondents generally expressed "high acceptance" and considered that their generation tended to be open "when it comes to gender equality issues." This sense of openness and broad acceptance was expressed by many respondents regardless of stage in medical program. Respondents also used phrases like "we don't resist", 'it's common in our generation", and "I accept and support" to convey their acceptance and openness toward LGBT individuals. Many respondents also mentioned they have peers who are publicly or openly LGBT, which also contributed to their self-perceived openness toward LGBT individuals (see Table 3A and 3B).

**Uncertainties in communication and sensitivity coated by acceptance on the surface.** Despite wide acceptance conveyed, respondents also expressed concerns about communicating with LGBT individuals and were still somewhat reluctant to discuss sensitive information, which they perceived as uncomfortable for patients. Thus, respondents mentioned they still "didn't know what ways [of communication] they should use so that they would not offend [LGBT individuals]" or "make them [LGBT patients] feel uncomfortable". Respondents believed they will need to "be more careful when communicating [with LGBT individuals]" or just "skip over more sensitive questions" in conversations (see Table 3C–3E).

One respondent even suggested that the broad acceptance and friendliness toward LGBT individuals could just be a surface phenomenon, given the evidence of bias respondents encountered during training (see Table 3F).

**Table 3. Themes, categories, and student quote examples.**

| **Attitude and Perceptions** |
| --- |

**Wide social acceptance and self-perceived openness** (28; 39% private, 61% public)

a. "We've seen a lot [of LGBT individuals] in our generation and are less likely to hold biases. On the contrary, the difference actually turns into our interest. [I have] encountered a lot of LGBT individuals in high school. It's not weird at all to see [LGBT individuals]." (School B: PGY, Focus Group)

b. "This [LGBT related] culture is already very common in my daily surrounding. I also had classmates who belong to this culture. Because of these encounters, I can understand and am less likely to resist or discriminate against (this culture)." (School B: Year 5, Interview)

**Uncertainties in communication and sensitivity coated by acceptance on the surface** (21; 71% private, 29% public)

c. "...Because we are just dealing with medical treatment, so we would not ask too many personal or sensitive questions. We would not ask what their private sexual life is like, except for in divisions like obstetrics or gynecology, which would more or less ask." (School A: Year 5, Interview)

d. "...Based on stereotypes, we considered these [LGBT] individuals to be in higher risk categories and don't let them donate blood. However, we need to be strategic in the way we communicate and not hurt them [LGBT individuals]" (School B: Year 4, Focus Group)

e. "...It's possible individuals know of [LGBT ] individuals but lack sufficient contact, they kind of know these groups of [LGBT] individuals, but do not know what ways to use so that they will not offend them [LGBT individuals]" (School B: Year 5, Focus Group)

f. "...But in reality, [we] are not very friendly [to LGBT individuals]. It is the atmosphere [social climate] that led everyone to think we are friendly, but even then, there would still be teachers saying [people] with AIDS must be youths or homosexual individuals." (School A: Year 4, Interview)

| **Bias and Stereotypes** |
| --- |

**Bias formed in training: Associating LGBT individuals with health risks and/or diseases** (56, 39% private, 61% public)

g. "It's quite obvious now that I think about it. For instance, the most typical [example] would be a homosexual patient. We would have more considerations about their risky sexual behaviors ...there are even doctors who teach us to make this judgement." (School A: Year 6, Interview)

**Bias in cognition: Reflexive thoughts & behaviors** (12, 8% private, 92% public)

h. "I don't think it's a bias, it's a statistical count. The probability of homosexual individuals getting HIV is indeed higher, so when we encounter homosexual individual, we would wonder whether these individuals contracted HIV. Throughout our medical training we have already been trained to see one thing and associate it with another. This is a reflexive behavior based on accumulated experiences." (School B: PGY, Focus Group)

i. "Teachers would provide examples (such as homosexual individuals might be at higher risk for sexually transmitted diseases) and across time we would memorize this example. Memorizing does not necessarily mean that's what you think is true, but when you try to recall the information, what you had memorized had actually become part of your thought process" (School B: Year 4, Interview)

| **Preparedness and Skillfulness** |
| --- |

**Treating everyone in the same way** (55; 45% private, 55% public)

j. "...We treated these [LGBT] patients the same way as we'd treat any patients. At the same time, however, they also have some risk factors. For instance, gays have higher risks for HIV based on objective evidence. Therefore, we would be more attentive to these aspects." (School B: Year 6, Interview)

**Prepared mentally, but lacking professional skills** (17; 41% private, 59% public)

k. "However, it is possible that people know a lot about [LGBT] individuals but lack actual interaction with them. Thus, they [college students] somewhat understand about the [LGBT] culture, but do not know how to interact with [LGBT individuals] without offending them." (School B: Year 5 –Focus Group)

**Sensitive topics? Only ask when necessary or just skip** (21; 71.4% private, 29% public)

l. "We would only ask this question [about sexual orientation] specifically when it pertains to sexually transmitted diseases." (School A: PGY, Focus Group)

m. "...I feel that once [we] enter clinical setting, our [preparedness] will rapidly increase because [we] will go verify whether what we think is right...[we] will combine our existing skills and what we had learned to quickly strengthen our abilities to interact with [LGBT patients]. I feel it's faster to enter the clinical stage and directly face [these patients]. However, it is also important to have some preparation before this stage." (School B: Year 4, Interview)

| **Curriculum** |
| --- |

**Adequacy of explicit curriculum** (43; 67% private, 33% public)

*(Continued)*

Table 3. (Continued)

| Attitude and Perceptions |
| --- |
| n. "There doesn't seem to be courses that teach us how to respond to these situations or about what kind of understanding we should have [about patients who are LGBT individuals]" (School A: Year 4, Interview) |
| **Implicit curriculum: Community participation and role modeling** (24; 42% private, 58% public) |
| o. "In terms of communication, I directly observe what attending physicians do. [They] rarely specifically tell us how we should communicate with these groups of people [LGBT individuals]. I think they don't intentionally identify these individuals. To them, patients are patients." (School B: PGY, Interview) |

Note. Total number of respondents who discussed each sub-theme and percentage of respondents from private/ public schools is included in parentheses.

Concern about empathic ways to communicate with LGBT individuals, and awareness of biases and stereotypes, were still largely present beneath the surface.

## Bias and stereotypes

Respondents observed that throughout training, they had heard about or experienced certain degree of stereotypes and biases in relation to LGBT individuals. However, they also recognized that biases and stereotypes might affect their reflexive thoughts and judgements.

**Bias formed in training: Associating LGBT individuals with health risks and/or diseases.** Some respondents reported an awareness that teachers might have bias against LGBT individuals. To facilitate student learning, some teachers incorporated in their teaching latent associations between LGBT status and health risks or diseases, which might prompt students to make these connections. They pointed out that teachers would "directly say male homosexual individual would get AIDS" (see Table 3G).

Many respondents reported they felt uncomfortable about these stigmas and had concerns that having biases or stigmas would affect their reasoning process. However, even with this awareness, this concern was still reflected in respondents who were in their later years of training.

**Bias in cognition: Reflexive thoughts & behaviors.** Despite respondents' awareness of bias against LGBT patients embedded in training, clinical stage respondents believed that associating LGBT status with health risks and diseases might not be considered as bias *per se*. They believed these associations were formed as byproducts of "doctors' own observations and experiences", and outcomes from previous studies (see Table 3H).

Respondents indicated that these associations would facilitate with making correct diagnoses. They also indicated these biases and stigmas were like "time-saving intuitions", "impressions based on life experiences", "an awareness or defensive mechanism," and were concerned that these biases would "gradually make [them] think it's right" and that they would "commit these biases to memory" (see Table 3I).

## Preparedness and skillfulness

**Treating everyone in the same way.** Respondents overall still saw themselves as adequately prepared to provide care for LGBT patients. Many respondents described LGBT individuals as being "no different from heterosexuals". Clinical respondents shared the same perception, equating non-discriminatory attitudes to having skills. Nevertheless, some respondents believed doctors should still pay special attention to and be more cautious when they encounter LGBT patients (see Table 3J).

**Prepared mentally, but lacking professional skills.** Results from previous study had shown most respondents rated their preparedness for taking care of LGBT patients (7.2/10) to be relatively higher than that for other patient demographic groups (3.7–6.6/10) [42]. A few respondents, however, noted they only rated their preparedness for taking care of LGBT patients to be higher if not considering "professional skills". Although some respondents indicated courses had provided opportunities to acquire understanding of and exposure to LGBT individuals, they raised concerns about actually interacting with these groups (see Table 3K).

Respondents determined their preparedness level by making associations with diseases and not discriminating against LGBT individuals, or communicating based on scientific evidence. Some indicated lack of professional skills and insufficient (clinical) exposure as factors that contributed to lower preparedness levels. This lack of exposure is particularly obvious when respondents discussed communicating with LGBT patients.

**Sensitive topics? only ask when necessary, or just skip.** Some clinical respondents suggested they would ask about patients' LGBT status only when they felt it was necessary or relevant for diagnosis, as this matter is "private." Only in circumstances relating to "sexually transmitted diseases", "obstetrics & gynecology", and "HIV & AIDS" should doctors ask about patients' sexual orientation. Moreover, some respondents reported they would avoid asking patients questions related to LGBT status as LGBT individuals would be reluctant to share this information since healthcare providers are not "part of the in-group." They articulated that "sensitive" questions should be skipped and "taboos" avoided in order to not make patients "angry" or "uncomfortable" (see Table 3L).

Some pre-clinical respondents believed they would make tremendous progress in preparedness once they reached the clinical stage of training (see Table 3M).

## Curriculum

**Adequacy of the explicit curriculum.** Respondents reported they had, to varying degrees, exposures to topics or information on LGBT individuals in the early years of their study (year one and two) and had a few courses that touched on LGBT-related topics. Some respondents provided examples of how teachers in general education courses, for instance, facilitated learning by inviting LGBT guest speakers to share their experiences that thus brought students awareness of different "medical perspectives". Nevertheless, such input to the curriculum was unsystematic. Just as our quantitative data [42] showed that inadequacy of cross-cultural training during medical school was a problem, respondents in interviews also showed concern for lack of training in topics related to LGBT individuals (see Table 3N).

Very few respondents reported having other relevant training in pre-clinical and clinical stages. Some clinical respondents stated they had encountered LGBT patients at clinical sites, but no further clarification was made except for biases observed in instructors' comments. Some respondents expressed there is still a gap in the curriculum that could be addressed by more in-depth discussions about how to interact with and care for LGBT patients.

**Implicit curriculum: Community participation and role modeling.** Respondents also mentioned elements of implicit curriculum as helpful in preparing students to work with LGBT patients. They cited informal learning opportunities including "observation of role models" and relevant activities such as student symposia and workshops on LGBT rights, the creation of an LGBT friendly campus environment, and exposure to friends or acquaintances who are LGBT. However, it should be noted that at the clinical stage, opportunities for respondents to engage with implicit curriculum beyond role modeling were absent from our data.

Some respondents expressed they observed what doctors do and learn from observations of "how the attending physician handles the situation and (we) learn from the experience." These

respondents believed "actually seeing" what their teachers and doctors do was more effective in helping them to prepare than just attending to lectures. However, partially echoing the quantitative data that showed students felt insufficient role modelling to be a problem [42], not all respondents had similar experiences since clinical learning or role modeling might vary (see Table 3O).

## Discussion

Our study is among the first in Asia to use qualitative approach to explore medical students'/trainees' perspectives about caring for LGBT patients, and their self-described preparedness to do so. The qualitative data has reflected some judgements, provided by student informants about their self-perceived competence and training in providing healthcare for LGBT patients. Overall, students expressed open attitudes that that reflects the changes in Taiwan's social climate toward SGM, yet ambivalence and uncertainties still remain not only in the ways to interact with LGBT patients and in their training, but also in making diagnosis. The results showed that the issues that surfaced in this study are related to bias formed in medical education, self-perceived preparedness versus actual competence, curriculum gap, and teachers as role models.

### Bias formed in medical education

Bias against LGBT individuals could be formed in the process of medical training and cultural stereotyping is a problem for doctors [42]. Implicit biases are also formed through the combination of cultural socialization and repeated experiences [43], and these biases might be difficult to unlearn once they become habitual [44]. Medical students and trainees from younger generations were aware that biases and stereotypes embedded in training might have negative influences on their judgements. For instance, indicative of students' awareness, stigmatization of HIV repeatedly emerged. A Taiwanese study also indicated some teachers may even express openly to students their hostilities toward LGBT individuals [45]. However, students also justified that having unconscious or conscious associations of bias with illness in their training environment is helpful. It is possible students may have unconsciously integrated biases and stereotypes related to LGBT individuals as part of their medical knowledge or training, rather than differentiating between the breadth of their medical knowledge and professional skills.

Examples most often mentioned by students were related to HIV, AIDS, and other sexually transmitted diseases. However, LGBT individuals are also at higher risks for certain cancers, experiences of discrimination and trauma within the healthcare system, mental health disorders, risk behaviors, and other health disparities [46,47]. Since respondents hardly mentioned these other issues, it is possible students were more influenced by common LGBT stereotypes than they were by deeper understanding of the epidemiologic patterns of disease and risk factors. This points to the need for additional education on caring for LGBT patients.

It is also evident that as students advanced through different stages of medical program, associations formed could be increasingly instilled into reflexive clinical reasoning processes, which might lead to the institutionalization of bias in medical culture [48]. This implies that the norm for medical training is to make a quick assumption of the most likely diagnosis based on clues of a patient's social background. However, the mindset of using this pattern of reasoning in medical practice may pose the risk of or even harden the stigmatization of diseases. This might result in medical practitioners becoming less sensitive to psycho-social concerns of patient-centered care, as well as possibilities for misdiagnosis or making assumptions that contribute to medical errors [49]. Medical practitioners' bias toward LGBT individuals might also further perpetuate physical and mental health disparities experienced by LGBT patients [50].

As forming associations between LGBT patients and diseases is not unique to Taiwan's medical students/trainees [49], medical education programs globally should consider the contexts in which these associations are formed and the impact they have on LGBT patients' health care to minimize these associations' negative influences on LGBT patients' access to adequate care.

## Self-perceived preparedness versus actual competence

Despite some mixed responses, most students nevertheless expressed strong confidence in their preparedness to care for LGBT patients and in their belief that they held non-discriminatory attitudes toward LGBT individuals as they objectively viewed scientific evidence as the basis for treating all patients. This mirrors similar findings from previous research [42], among Harvard medical students [51] and among residents in a national survey [52] who felt well-prepared, but were less confident in their specific skills of caring for LGBT patients. The findings support the idea that social changes in Taiwan have had a significant impact on the students' attitudes toward LGBT patients. Exposures to social movements and LGBT individuals have evidently helped to increase understanding of and familiarity with these groups. Whereas students considered themselves as being prepared to care for LGBT patients, very few of them could actually articulate in detail the nature of their competence. They tended to equate non-discriminatory attitude as being sufficiently prepared, while their skills were shown to be inadequate when it came to real interactions with LGBT patients. It also did not occur to most students that biopsychosocial knowledge (such as traumatization by discrimination) and communication skills are equally essential for developing mutual trust in physician-patient relationships [53] and thus providing appropriate care. It is evident that students were not sure where to draw the line between asking questions that were "too sensitive" and those that were "necessary" because they might affect provision of effective care [54,55]. The incongruence in perceived and actual preparedness should be considered across different layers of Taiwan's social environment and medical education curriculum.

## Curriculum gap

Results suggest medical education in Taiwan has not evolved along with nor responded sufficiently to social changes and diverse patient needs. Despite legal advances in LGBT rights and requirement for medical education programs to include training to eliminate gender biases, LGBT related issues have not been well addressed in the curricular content of Taiwanese medical institutions. Other countries are also dealing with similar curricular gaps [29,32,35,56,57]. It is apparent that the curricular content lacks consistent and systematic design across formal, informal, and hidden curriculum when it comes to LGBT related issues, yet students' views of factors that contribute to their preparedness level appear to be tied to medical education curriculum. There are still gaps in formal curriculum regarding the training stage in which relevant courses are taught. Although a few courses in the earlier stages of medical education were mentioned, it was apparent that the aforementioned criteria stated by TMAC [40] and LCME [41] still need to be more systematically and fully integrated in Taiwan's medical education curriculum.

Additionally, teachers' cultural and gender sensitivity could play a part in the hidden curriculum, and conscious or unconscious biases and stereotypes embedded in teaching remain a challenge to change in the educational climate [45,58–60]. Further faculty development will be necessary to address these issues, to identify barriers to inclusion of LGBT care competency, and to design and implement effective curricula [56].

Some students believed they would 'naturally' and 'quickly' acquire the skills once they reach clinical stage, but in an earlier study of clinical students in a psychiatry rotation, students

still have limited understanding of how to engage with transgender patients despite having reached the clinical setting [38]. The intended learning outcomes of relevant training at the clinical stage appears to be unclear and may be largely affected by clinicians and their role modeling behavior, and the hidden curriculum in clinical environment.

### Teachers as role models

Although teachers are important role models, messages that emerged about their influence are paradoxical. Students were aware of biases and stereotypes that exist in teaching; however, observing clinical teachers' communication skills and behaviors was seen as an effective way of learning. If clinical teachers associated certain health habits with particular cultural groups, students might reflexively or subconsciously include these behaviors in their understanding of diseases. Clinical observation is important, but whether students have adequate role modeling is not assured. Teachers' cultural competence warrants caution and requires further faculty training, particularly if they were seen as role models in students' development of professionalism [60–62]. Since current health care providers in Taiwan were trained when LGBT issues were less openly discussed, it follows that the informal learning environments both on campus and in clinical affiliates may be lacking in quality mentoring [60–62]. Similar concerns were expressed in western countries such as the US as many clinical educators also expressed receiving little training on LGBTQ+ health themselves or ways to teach these contents [63].

The degree to which the curricular content has served to eliminate health disparities among LGBT patients appears to be insufficient. Medical educational program directors and educators need to identify and address the gap between expected competence, students' self-assessed competence, and the curricula throughout the various stages of students' medical education.

### Limitations

This study has several limitations in terms of representativeness. We conducted qualitative research in only two medical schools, which may limit the generalizability of findings. It is possible that students who held different views or had greater knowledge about LGBT patients did not participate. However, there is reason to be confident that the views expressed by respondents are typical of those held by students of different Taiwan medical schools. First, the diversity of students in this study may be limited by uniform selection criteria adopted by medical schools across Taiwan. Nevertheless, uniform selection criteria also make these perspectives more generalizable to other medical students. Thus, despite the different curriculum design and duration, students of both schools had similar perceptions. Participants' actual knowledge of specific medical services that LGBT patients might need was not assessed. Since this study was cross-sectional, cohorts of students were not followed as they advanced in medical training. Thus, the results may be confounded by variations in training stages. Participants might have also withheld their own opinions out of fear of repercussion or disapproval from peers. These concerns will require further consideration in future studies.

### Conclusion

This study found students and trainees had higher acceptance of LGBT individuals, a finding that can mostly be attributed to the social climate, frequent exposure to LGBT-related issues, personal experiences, and to a lesser extent, social desirability. However, despite that social, cultural and policy changes have pushed Taiwan's social climate to align with more acceptive attitudes observed in some Western cultures, it is apparent that these shifts have not been followed by more systematically structured curriculum as bias against LGBT individuals and stigmatization of diseases are still present in medical education program, and it is likely that quick

association formed between diseases and stereotypes will be transformed into a basis for their knowledge and patterns of clinical reasoning. Therefore, medical education programs need to carefully address issues ranging from curricular content to continuous professional and faculty development, and design an integrated longitudinal formal curriculum that extends to the explicit and implicit curricula at clinical sites. Teachers also require continuous training as role models who are capable of providing an unbiased curriculum and of helping students to learn to provide care that meets the needs of diverse patient populations.

This study sheds light on the lag between social changes and medical education in Taiwan, an East Asian country that is at the forefront of embracing gender diversity. Although this study focused on medical students/trainees in Taiwan, findings from this study raised several important questions that are also encountered by medical education programs globally as they reform and design more structured curriculum to fit their local and national contexts. Thus, these results can serve as a point of reference for medical educators in similar social contexts and help them to reconsider their own medical curricula to prepare future physicians to care for more diverse patient population while also empowering them to engage in improving future healthcare services. Essentially, this study fills a gap in the emerging world map of diversity education in medical education.

## Supporting information

**S1 File. Nvivo codebook.**
(DOCX)

## Acknowledgments

The authors wish to thank Dr. John Corbett and Dr. Kwong D. Chan for reading the article and giving critical feedback.

## Author Contributions

**Conceptualization:** Peih-Ying Lu.

**Data curation:** Peih-Ying Lu, Anna Shan Chun Hsu.

**Formal analysis:** Anna Shan Chun Hsu.

**Funding acquisition:** Peih-Ying Lu.

**Investigation:** Peih-Ying Lu.

**Methodology:** Peih-Ying Lu.

**Project administration:** Peih-Ying Lu.

**Software:** Anna Shan Chun Hsu.

**Supervision:** Peih-Ying Lu.

**Validation:** Peih-Ying Lu.

**Writing – original draft:** Peih-Ying Lu, Anna Shan Chun Hsu.

**Writing – review & editing:** Peih-Ying Lu, Anna Shan Chun Hsu, Alexander Green, Jer-Chia Tsai.

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
