## [Decision Letter · Decision Letter 0]

21 Jan 2022

PONE-D-21-38628Medical students’ perceptions of their preparedness to care for LGBT patients in Taiwan: Is medical education keeping up with social progress?PLOS ONE

Dear Dr. Lu,

Thank you for submitting your manuscript to PLOS ONE. After careful consideration, we feel that it has merit but does not fully meet PLOS ONE’s publication criteria as it currently stands. Therefore, we invite you to submit a revised version of the manuscript that addresses the points raised during the review process.

We look forward to receiving your revised manuscript.

Kind regards,

Haoran Xie

Academic Editor

PLOS ONE

Journal Requirements:

Reviewers' comments:

Reviewer's Responses to Questions

**Comments to the Author**

1. Is the manuscript technically sound, and do the data support the conclusions?

Reviewer #1: No

Reviewer #2: Yes

Reviewer #3: Yes

2. Has the statistical analysis been performed appropriately and rigorously? 

Reviewer #1: No

Reviewer #2: I Don't Know

Reviewer #3: Yes

3. Have the authors made all data underlying the findings in their manuscript fully available?

Reviewer #1: No

Reviewer #2: Yes

Reviewer #3: Yes

4. Is the manuscript presented in an intelligible fashion and written in standard English?

Reviewer #1: No

Reviewer #2: No

Reviewer #3: Yes

5. Review Comments to the Author

Reviewer #1: I have reviewed the paper by Lu et al . on Medical students’s perception of readiness for LGTB patients.

Overall, although health equity may be affected for the LGBT community due to discrimination, this does not necessarily goes hand with the care provided by health care professionals. So statements like “…health care professionals to engage with a population of LGBT individuals who are open to revealing their sexual…” , what is engagement referring to/ Health care should be provided blindly to whomever needs it.

The Introduction needs some evidence that health providers are not treating LGBT patients the same way, to justify this kind of study. References 20-22 do not provide such evidence. If they are unable to provide such evidence, the Introduction should then start on line 51.

The statement “… due to the fact that Asian countries are still relatively conservative about topics related to LGBT individuals in medical education [24].” is an overstatement. Is this truly a fact? If so please provide evidence beyond reference 24.

Introduction. I am not sure that ‘prevalent” is the best word to describe the movements to ensure equity of health.

Last part of the Introduction mentions that “Findings from quantitative data collected by the current research team showed student respondents felt somewhat adequately prepared to provide care for LGBT patients while teachers felt the contrary [36].” This discrepancy between students and teachers is a major finding.

I would like to see some sort of trend analysis with the current study across the years of medical school.

Sampling 9to reach the 89 students for focus groups was done randomly. I am assuming randomly after categorization, as Table 1 shows a pretty comparable distribution by gender and year in medical school. There is no description of the characteristics of the two medical schools involved.

Table 3 is supposed to show results. But they are not even percentages or proportions for the different categories. Despite this, we see statements like “Respondents overall still saw themselves a 211 s adequately prepared to provide care for LGBT patients.” Or ‘A few respondents, however, noted they only rated their preparedness for taking care of LGBT patients to be higher if not considering “professional skills”.”, “Some clinical respondents suggested they would…”etc.

There are some sentences that need grammatical revision.

Reviewer #2: Dear Authors

It is a topic that could be needed in medical education. You have messed up methods, and results it is very hard to understand what the methodology / results. Abstract: Introduction: Integrating training on health equity of sexual and gender minorities (SGM) in medical education has been challenging globally despite emphasis on the need for medical students to develop competence to provide adequate care for diverse patient groups. This study elicits Taiwanese medical students’ perceptions of their values and preparedness to care for LGBT patients, using a qualitative approach..

What is LGBT ( Lesbian Gay ........... ) the first time you should write full

Eighty-nine medical students/trainees from two Taiwanese medical schools 10 participated in focus groups and individual interviews…

How many Focus groups (n= ) and interviewee (n= )

Results: Participants (i) expressed wide social acceptance and openness toward LGBT individuals, but were unsure of ways to communicate with LGBT patients; (ii) confirmed that stigmatization and biases might be developed during their training; (iii) recognized gender stereotypes could have negative impacts on clinical reasoning; (iv) considered themselves prepared to care for LGBT patients, yet equated non-discriminatory attitudes to preparedness; (v) acknowledged a lack of relevant professional skills; ; (vi) implicated curriculum did not address LGBT issues systematically and explicitly.

There is no variations in expressions.

Reviewer #3: I liked the idea behind this paper because it strives to explore fundamental questions of the students preparedness for care as a physician in the practices. Making a novel contribution in the area is a challenge. This study would be an addition to the existing literature.

The methodology section addressed key aspects. The research paradigm was explicitly identified, and the authors did provide sufficient analysis of each theme.

6. PLOS authors have the option to publish the peer review history of their article (what does this mean?). If published, this will include your full peer review and any attached files.

Reviewer #1: **Yes: **Jorge L. Cervantes

Reviewer #2: No

Reviewer #3: No

---

## [Author Response · Author response to Decision Letter 0]

3 Mar 2022

Dear Editor and Reviewers,

Thank you very much for reviewing our manuscript and for your valuable comments. The page and line numbers below are listed according to the “Revised Manuscript with Track Changes” file. We would like to give responses to all the comments point by point as follows: 

Reviewer #1: I have reviewed the paper by Lu et al. on Medical students’s perception of readiness for LGTB patients.

Author’s Response (AR): We thank you for reviewing our paper and providing comments to help us improve the quality of our manuscript. 

1. Overall, although health equity may be affected for the LGBT community due to discrimination, this does not necessarily goes hand with the care provided by health care professionals. So statements like “…health care professionals to engage with a population of LGBT individuals who are open to revealing their sexual…” , what is engagement referring to/ Health care should be provided blindly to whomever needs it.

AR: We agree that health care should be provided to any individual who needs it without discrimination. This is also one of our purposes for this study (i.e. to provide more insights and understanding of that topic in current medical education). 

“Engage” or “engagement” in this context refers to the thought that with shifts in Taiwan toward more acceptance and open attitudes toward LGBT individuals, health care professionals are now attending to or caring for more LGBT patients who are willing to reveal their sexual orientation or gender identities (whereas in the past, these patients might be more hesitant to reveal this information when being attended to by health care professionals). To avoid further confusion in what we meant by “engage with”, we changed the phrase to “care for” instead (pg.3, line 50).

2. The Introduction needs some evidence that health providers are not treating LGBT patients the same way, to justify this kind of study. References 20-22 do not provide such evidence. If they are unable to provide such evidence, the Introduction should then start on line 51.

AR: We are unsure whether the reviewer meant references 20-22 or was actually referring to references 10-12. References 10-12 pertain to attitudes factors that might influence health equity of LGBT individuals in Taiwan and other east Asian countries while references 20-22 provide context for how bias and inadequate training might contribute to health disparities. We added a few more references that reflect disparities in care for LGBT patients:

“Across different age groups, patients who identify as Lesbian, Gay, Bisexual, or Transgender (LGBT) often face challenges and disparities when seeking to satisfy their health care needs [4-8].” (pg. 3, lines 32-34)

3. The statement “… due to the fact that Asian countries are still relatively conservative about topics related to LGBT individuals in medical education [24].” is an overstatement. Is this truly a fact? If so please provide evidence beyond reference 24.

AR: We modified this statement to not overstate the point, and we also added several references to support it (pg. 4, lines 64-67). Although Taiwan has been deemed socially as “the most open society toward LGBTQ in East Asia, other countries like China and Singapore had little change in their level of acceptance toward LGBTQ community (Zhou, 2020). 

Moreover, less research has been conducted on topics related to LGBT individuals in medical education in Asian countries. For instance, Yamazaki et al. (2020) is one of very limited studies we found in Asia that attempted to survey undergraduate medical education programs about the inclusion of LGBT-related contents in undergraduate medical education in Japan. Compared with Obedin-Maliver et al’s study (2011), which explored similar topics in medical curricula in the US, response rate in Yamazaki et al. was only 46% (22/80) vs Obedin-Maliver et al.’s response rate of 85.2% (150/176). Authors from both studies suggested there are still room for improvement in increasing LGBT-related curricula. In another study conducted in South Korea (Lee et al., 2021), the authors mentioned “the topic of LGBT care remains excluded from medical curricula throughout Korea, and there has also been zero research on transgenderism among students in the country’s medical profession.” We also added this reference to our paper. (pg. 4, line 67)

4. Introduction. I am not sure that ‘prevalent” is the best word to describe the movements to ensure equity of health.

AR: In this context, we used the word “prevalent” to indicate that there had been more movements to ensure equity of health care provision for LGBT patients in Western societies than in Asian ones. Thus, we changed the phrasing to read: “Over the past few decades, movement to ensure equity of health care provision for LGBT patients has been more common in Western societies…” (pg. 3, line 37)

5. Last part of the Introduction mentions that “Findings from quantitative data collected by the current research team showed student respondents felt somewhat adequately prepared to provide care for LGBT patients while teachers felt the contrary [36].” This discrepancy between students and teachers is a major finding. I would like to see some sort of trend analysis with the current study across the years of medical school.

AR: We thank the reviewer for the suggestion to conduct trend analysis. Trends across years of medical school are also something we are interested in assessing and will plan to collect more data to add to our preliminary analyses. Since the focus of our manuscript is primarily on qualitative data and analyses, a trend analysis would not be feasible in this paper. 

6. Sampling 9to reach the 89 students for focus groups was done randomly. I am assuming randomly after categorization, as Table 1 shows a pretty comparable distribution by gender and year in medical school. There is no description of the characteristics of the two medical schools involved.

AR: We want to clarify that the reviewer is referring to Table 2 rather than Table 1. Sampling of focus groups was done randomly after categorization of year of study to ensure that we would at least have one group of students per cohort. However, gender across focus groups was by random. 

On pg. 5-6, lines 100-103 and in the Note section of Table 2, we described the compositions of medical programs in Taiwan. The description is applicable to the majority of medical schools in Taiwan, including the two that are included in the study. These descriptions are listed below:

“In Taiwan, most medical schools are undergraduate entry level, with years one to four as the pre-medical and pre-clinical stages and years five to six as the clinical stage. All medical graduates attend the compulsory two-year postgraduate (PGY) general medicine training program.” (pg. 5, lines 97-100)

“Taiwanese medical schools generally admit high school graduates. Starting from 2013, the original 7-year medical education program changed to 6-years, with the original final internship year moved to postgraduate clinical training (PGY). As the study was conducted in 2015-2017 before the implementation of the two-year PGY program, only the first PGY year was included in the study.” (Table 2)

In addition, we added the phrase “one public and one private” and the word “southern” to provide additional descriptions: 

“…was administered to 1545 medical students from one public and one private medical schools in southern Taiwan…” (pg. 6, lines 102-103)

7. Table 3 is supposed to show results. But they are not even percentages or proportions for the different categories. Despite this, we see statements like “Respondents overall still saw themselves a 211 s adequately prepared to provide care for LGBT patients.” Or ‘A few respondents, however, noted they only rated their preparedness for taking care of LGBT patients to be higher if not considering “professional skills”.”, “Some clinical respondents suggested they would…”etc.

AR: We added the total number of respondents included in discussions of each subtheme in parentheses in Table 3. 

8. There are some sentences that need grammatical revision.

AR: Thank you for bringing this to our attention. We went through the manuscript again to make some revisions. 

Reviewer #2: Dear Authors

1. It is a topic that could be needed in medical education. You have messed up methods, and results it is very hard to understand what the methodology / results. 

AR: We thank the reviewer for recognizing the importance of this topic in medical education. We went through the methods and results sections to try to re-organize and clarify these sections accordingly. Below is a breakdown of how we structured the methods and results sections. 

The methods section is split into three sections: participants, design of study, and analysis section: 

The participants section first starts with a brief description of medical education program in Taiwan to provide basic background information for the sample of students, followed by short explanation of how participants in this study were selected from the pool of students who completed the quantitative survey in the first phase of our study. We presented Table 1 just to provide some insight to the diverse cross-cultural groups that students were asked about in the first phase. Demographics for the 89 students who participated in focus groups and interviews are presented in Table 2. 

The design section focuses primarily on the second (qualitative) phase of our study, which is also the main focus of this manuscript. We presented a description of the process in which we conducted the focus groups and interviews, ethics approval information, questions asked (box 1). 

The analysis section describes the process in which we conducted qualitative analysis. 

The results section presents Table 3 (listing out the four themes, nine categories or sub-themes, and example quotes from students). The themes and categories are then presented.

2. Abstract: Introduction: Integrating training on health equity of sexual and gender minorities (SGM) in medical education has been challenging globally despite emphasis on the need for medical students to develop competence to provide adequate care for diverse patient groups. This study elicits Taiwanese medical students’ perceptions of their values and preparedness to care for LGBT patients, using a qualitative approach..

What is LGBT ( Lesbian Gay ........... ) the first time you should write full

AR: Per reviewer’s suggestion, we added the full term for the acronym in the abstract the first time we mentioned it (pg. 2 Abstract, line 7; pg. 3 Introduction, line 32-33)

3. Eighty-nine medical students/trainees from two Taiwanese medical schools participated in focus groups and individual interviews…

How many Focus groups (n= ) and interviewee (n= )

AR: Per reviewer’s suggestion, we added this information in our abstract. (pg. 2 Abstract, lines 11-12)

4. Results: Participants (i) expressed wide social acceptance and openness toward LGBT individuals, but were unsure of ways to communicate with LGBT patients; (ii) confirmed that stigmatization and biases might be developed during their training; (iii) recognized gender stereotypes could have negative impacts on clinical reasoning; (iv) considered themselves prepared to care for LGBT patients, yet equated non-discriminatory attitudes to preparedness; (v) acknowledged a lack of relevant professional skills; ; (vi) implicated curriculum did not address LGBT issues systematically and explicitly.

There is no variations in expressions.

AR: We are not sure what the reviewer meant by “there is no variations in expressions”, however, we did purposely phrase the expressions syntactically similar to keep them consistent across the phrases. 

Reviewer #3: I liked the idea behind this paper because it strives to explore fundamental questions of the students preparedness for care as a physician in the practices. Making a novel contribution in the area is a challenge. This study would be an addition to the existing literature.

The methodology section addressed key aspects. The research paradigm was explicitly identified, and the authors did provide sufficient analysis of each theme.

AR: We thank Reviewer 3 for reviewing our manuscript and for recognizing the fundamental questions we addressed as well as the contribution our paper would add to the existing literature.

Journal Requirements:

AR: We thank the journal for the reminder and have double-checked to ensure our manuscript meets the requirements. 

AR: We have added additional details along with our ethics statement in the Methods section. The following information is added:

“Ethics approval for this study was obtained from the National Cheng Kung University Governance Framework for Human Research Ethics (Approval No.: NCKU HREC-E-106-291-2/ NCKU HREC-E-104-035-2). Participants in focus groups and interviews provided written consent at the beginning of the sessions.” (pg. 8, lines 140-143)

AR: We have moved the ethics statement to the Methods section of our manuscript. (pg. 8, lines 140-142).

---

## [Decision Letter · Decision Letter 1]

24 May 2022

PONE-D-21-38628R1Medical students’ perceptions of their preparedness to care for LGBT patients in Taiwan: Is medical education keeping up with social progress?PLOS ONE

Dear Dr. Lu,

Thank you for submitting your manuscript to PLOS ONE. After careful consideration, we feel that it has merit but does not fully meet PLOS ONE’s publication criteria as it currently stands. Therefore, we invite you to submit a revised version of the manuscript that addresses the points raised during the review process.

We look forward to receiving your revised manuscript.

Kind regards,

Haoran Xie

Academic Editor

PLOS ONE

Reviewers' comments:

Reviewer's Responses to Questions

**Comments to the Author**

1. If the authors have adequately addressed your comments raised in a previous round of review and you feel that this manuscript is now acceptable for publication, you may indicate that here to bypass the “Comments to the Author” section, enter your conflict of interest statement in the “Confidential to Editor” section, and submit your "Accept" recommendation.

Reviewer #1: (No Response)

Reviewer #2: All comments have been addressed

2. Is the manuscript technically sound, and do the data support the conclusions?

Reviewer #1: No

Reviewer #2: Yes

3. Has the statistical analysis been performed appropriately and rigorously? 

Reviewer #1: No

Reviewer #2: Yes

4. Have the authors made all data underlying the findings in their manuscript fully available?

Reviewer #1: No

Reviewer #2: Yes

5. Is the manuscript presented in an intelligible fashion and written in standard English?

Reviewer #1: No

Reviewer #2: Yes

6. Review Comments to the Author

Reviewer #1: Response to my comments (6 and 7) are non-satisfactory.

They need to present clear categories with percentages in table 3 and cross tabulate against the National and the Private schools (using a Fisher or a Chi square test at leas).

The Results, as they are listed in the Abstract are unsupported results if no analysis is done.

Reviewer #2: Authors have complied all the issues raised earlier

one minor correction they should do

Introduction - Line 80 (Page 15)

The current study is part of a larger project that used mixed methods to examine Taiwan medical students’ perspectives in caring for patients from diverse groups

Comment: is a part of larger project that used mixed method should be delete, it is confusing , better to say ‘qualitative’ method was employed ............

7. PLOS authors have the option to publish the peer review history of their article (what does this mean?). If published, this will include your full peer review and any attached files.

Reviewer #1: **Yes: **Jorge Cervantes

Reviewer #2: No

---

## [Author Response · Author response to Decision Letter 1]

2 Jun 2022

Please see below (and uploaded file) to see our responses to reviewers' comments:

Dear Editor and Reviewers,

Thank you for reviewing our revised manuscript again and for providing further comments. The page and line numbers below are listed according to the “Revised Manuscript with Track Changes” file. We would like to respond to all the comments point by point as follows: 

Comments to the Author

1. If the authors have adequately addressed your comments raised in a previous round of review and you feel that this manuscript is now acceptable for publication, you may indicate that here to bypass the “Comments to the Author” section, enter your conflict of interest statement in the “Confidential to Editor” section, and submit your "Accept" recommendation.

Reviewer #1: (No Response)

Reviewer #2: All comments have been addressed

Author’s Response (AR): We thank Reviewers 1 & 2 for reviewing our paper again and for providing additional comments. We will address these comments below. 

2. Is the manuscript technically sound, and do the data support the conclusions?

Reviewer #1: No

Reviewer #2: Yes

AR: Since the two reviewers provided different responses, we are unsure which part(s) Reviewer 1 felt did not meet these criteria. We felt our study was conducted rigorously with appropriate sample size for qualitative study. Our focus groups and interviews were properly conducted, transcribed, and analyzed. Our conclusions were also drawn from the data we had collected and presented. 

3. Has the statistical analysis been performed appropriately and rigorously?

Reviewer #1: No

Reviewer #2: Yes

AR: Similar to our response above, we are unsure which part(s) of our statistical analysis Reviewer 1 felt was not done appropriately and rigorously. We provided demographics of our participants and added in the number of respondents as well as percentages in Table 3 as Reviewer 1 had suggested. As the rest of our analysis are qualitative, additional quantitative analysis would not be feasible. 

4. Have the authors made all data underlying the findings in their manuscript fully available?

Reviewer #1: No

Reviewer #2: Yes

AR: As our data are qualitative transcriptions and contain quotes provided by student respondents, including the full transcriptions might be problematic. Participants were not informed, nor did they provide approval for publication of their full transcriptions. Our transcriptions could also include information that would make the participants identifiable. Thus, we have made our NVIVO codebook for focus groups and interviews available instead as this provide readers complete information of how we coded our transcriptions. 

5. Is the manuscript presented in an intelligible fashion and written in standard English?

Reviewer #1: No

Reviewer #2: Yes

AR: Similar to previous questions, we are unsure which part(s) Reviewer 1 thought was/were written in unintelligible fashion or were not written in standard English. We went through the manuscript and to the best of our abilities, made minor revisions to language that might be unclear. Among the four researchers, one is an English native speaker, two Chinese and English bilingual speakers (one received complete education and one received master and PhD degrees in English speaking countries). We also asked one native English- speaking scholar to read through the paper again to help ensure the manuscript is intelligible and written in standard English. 

6. Review Comments to the Author

Reviewer #1: Response to my comments (6 and 7) are non-satisfactory.

They need to present clear categories with percentages in table 3 and cross tabulate against the National and the Private schools (using a Fisher or a Chi square test at leas).

The Results, as they are listed in the Abstract are unsupported results if no analysis is done.

AR: Reviewer 1’s previous comments were regarding the provision of descriptions of the two medical schools involved (comment 6) and results showing percentages for the different categories (comment 7). 

In our previous response, we confirmed that sampling of focus groups was done randomly after categorization of years of study to ensure that we would at least have one group of students per cohort. However, gender across focus groups was by random. We also added descriptions of the two medical schools involved as follows:

On pg. 5-6, lines 100-103 and in the Note section of Table 2, we described the compositions of medical programs in Taiwan. The description is applicable to the majority of medical schools in Taiwan, including the two that are included in the study. These descriptions are listed below:

“In Taiwan, most medical schools are undergraduate entry level, with years one to four as the pre-medical and pre-clinical stages and years five to six as the clinical stage. All medical graduates attend the compulsory two-year postgraduate (PGY) general medicine training program.” (pg. 5, lines 97-100)

“Taiwanese medical schools generally admit high school graduates. Starting from 2013, the original 7-year medical education program changed to 6-years, with the original final internship year moved to postgraduate clinical training (PGY). As the study was conducted in 2015-2017 before the implementation of the two-year PGY program, only the first PGY year was included in the study.” (Table 2)

In addition, we added the phrase “one public and one private” and the word “southern” to provide additional descriptions: 

“…was administered to 1545 medical students from one public and one private medical schools in southern Taiwan…” (pg. 6, lines 102-103)

Also, as Taiwan’s medical schools (both private and public) adopt uniform selection criteria, the demographics of students at the two schools are relatively similar. 

Per the reviewer’s suggestion, we also added the total number of respondents included in discussions of each category in parentheses. To provide a clearer presentation, we have bolded the text for the categories and in addition, added percentages to tabulate the distributions of quotations from public and private schools.

However, as the purpose of our study is to explore students’ qualitative responses regarding their perceptions and not to compare quantitatively between the proportion of responses from public and private schools, we feel that including a tabulation would not contribute substantial meaning to help readers understand the results and might detract readers’ attention from the study’s main results. Nevertheless, we added these percentages to Table 3. 

Reviewer #2: Authors have complied all the issues raised earlier

one minor correction they should do

Introduction - Line 80 (Page 15)

The current study is part of a larger project that used mixed methods to examine Taiwan medical students’ perspectives in caring for patients from diverse groups

Comment: is a part of larger project that used mixed method should be delete, it is confusing , better to say ‘qualitative’ method was employed ............

AR: We changed introduction (pg. 15, line 80) to say the following:

“The current study used qualitative method to examine medical students’ preparedness to care for LGBT patients. This study is also part of a larger project that used mixed methods to examine…” 

In addition to clarifying that the current study employs qualitative method, we felt we should still inform readers that this study originally stemmed from a larger study, which employed mixed methods. Readers can thus refer to our previous study if necessary. Thus, we revised the sentence as shown above. 

---

## [Editor Report · Decision Letter 2]

20 Jun 2022

Medical students’ perceptions of their preparedness to care for LGBT patients in Taiwan: Is medical education keeping up with social progress?

PONE-D-21-38628R2

Dear Dr. Lu,

We’re pleased to inform you that your manuscript has been judged scientifically suitable for publication and will be formally accepted for publication once it meets all outstanding technical requirements.

Kind regards,

Haoran Xie

Academic Editor

PLOS ONE

---

## [Editor Report · Acceptance letter]

27 Jun 2022

PONE-D-21-38628R2 

Medical students’ perceptions of their preparedness to care for LGBT patients in Taiwan: Is medical education keeping up with social progress? 

Dear Dr. Lu:

I'm pleased to inform you that your manuscript has been deemed suitable for publication in PLOS ONE. Congratulations! Your manuscript is now with our production department. 

Kind regards, 

on behalf of

Professor Haoran Xie 

Academic Editor

PLOS ONE